# The Utility of ABO Testing in Pediatric Patients Undergoing Elective Surgery

**DOI:** 10.3390/jcm8091372

**Published:** 2019-09-02

**Authors:** Min Seob Kim, Sanghyun Ahn, Eun-Jin Chung, Seungeun Choi, Jin-Tae Kim, Ho-Geol Ryu

**Affiliations:** 1Department of Surgery, Seoul National University Hospital, Seoul 03080, Korea; 2Department of Anesthesiology, Seoul National University Hospital, Seoul 03080, Korea

**Keywords:** ABO/Rh, transfusion, pediatric anesthesia

## Abstract

Background: Patients for elective operation often undergo routine ABO/Rh type and screening test for potential need of transfusion. Some institutions require double verification of ABO/Rh type. We evaluated the clinical practice pattern of performing ABO/Rh type and screening test in pediatric patients undergoing elective operation. Methods: Electronic medical records from pediatric patients who underwent elective surgery between June 2006 and June 2010 were retrieved. The frequency of ABO/Rh type and screening test and the incidence of packed red blood cell (pRBC) request and pRBC dispatch from the blood bank among those tested were analyzed by year and the surgical department. Results: Of the 23,631 patients, the incidence of ABO/Rh type and screening was 32.2%, and pRBC was dispatched in 37.9% of these patients. The incidence of ABO/Rh type and screening varied between 1.5% and 97.9% among surgical departments and also within the surgical departments depending on the type of surgery. The incidence of ABO/Rh type and screening showed a decreasing trend over the study period. Conclusions: There was significant variability among and within the surgical departments in the incidence of ABO/Rh type and screening in children undergoing elective surgery. A tailored approach may be beneficial to the patient in terms of comfort and cost.

## 1. Introduction

ABO/Rh type and screening is performed to prevent transfusion complications from incompatibility and other irregular antibodies [1] Preoperative ABO/Rh type and screening is often routinely performed in patients scheduled to undergo elective surgery with a potential need of transfusion [2,3].

In the pediatric population, especially in younger children, drawing blood can be stressful not only for the child, but also for parents and the provider performing the blood draw. Legal circumstances in Korea led to hospital policies that only allow physicians to draw the blood sample for ABO/Rh type and screening due to serious sentinel events tied to sampling/labelling error. As a result, when ABO/Rh type and screening is ordered with other laboratory tests, the patient often has to be sampled twice. Moreover, for patients whose ABO/Rh type has not been tested previously, many institutions require crosschecking with two separate samples to ensure accuracy.

Most elective pediatric surgical procedures do not require perioperative transfusion [4]. Therefore, preoperative ABO/Rh type and screening should be performed only if the transfusion is expected strongly during surgery, taking into account the stress of the patient’s blood draw as well. Evaluation of the probability of blood transfusion for frequently performed surgical procedures is essential in order to reduce the number of unnecessary preoperative ABO/Rh type and screening. To this end, we evaluated the current practice of perioperative blood transfusion in the largest pediatric hospital in Korea.

## 2. Methods

Seoul National University Children’s Hospital is the largest children’s hospital in Korea, with 312 beds and an average of about 1000 daily outpatient visits. The study protocol was approved (H-1007-215-326) and the requirement for written informed consent was waived by the institutional review board of the hospital. Electronic medical records of Seoul National University Children’s Hospital were retrieved and analyzed for the study.

### 2.1. Patient Population

Patients who were 15 years old or younger at the time of surgery undergoing elective surgery in Seoul National University Children’s Hospital between June 2006 and June 2010 were analyzed. Emergency operations were excluded.

### 2.2. Data Acquisition

Variables including surgical department, type of surgery, ABO/Rh type, and screening test within 3 days prior to surgery, requests for packed red blood cells (pRBC) on the day of surgery, and dispatch of pRBCs to the operating theatre on the day of surgery were extracted and recorded. Given the policy of the hospital blood bank that only allows crossmatching with blood that has been sampled within 3 days, ABO/Rh type and screening within 3 days of their surgery strongly suggests that the test was performed to prepare for potential transfusion during surgery.

### 2.3. Statistical Analysis

The following ratios were analyzed: patients with ABO/Rh type and screening test among all patients, patients for whom pRBC was requested among patients with ABO/Rh type and screening test**,** patients for whom the requested pRBC was dispatched to the operating theater among patients for whom pRBC was requested, crossmatch to transfusion ratio, and patients for whom the requested pRBC was dispatched to the operating theater among patients with ABO/Rh type and screening test.

The ratios mentioned above were calculated for the three most frequently performed surgeries of each surgical department. In addition, the ten most frequently performed surgeries overall were analyzed and summarized by year. Ratios were compared with either Pearson’s chi-square test or Fischer’s exact test using Stata 10.0 IC (College Station, TX, USA). A *p*-value of <0.05 was considered statistically significant.

## 3. Results

We obtained 23,631 patients’ records from pediatric patients who underwent elective surgery from June 2006 to June 2010. The overall incidence of ABO/Rh type and screening test for pediatric patients undergoing elective surgery was 32.2%, and pRBC was only dispatched in 37.9% of these patients.

Table 1 shows the incidence of ABO/Rh type and screening from eight surgical departments. ABO/Rh type and screening was performed most frequently in the department of cardiothoracic surgery and most infrequently in the ophthalmology department (97.9% vs. 1.5%). The cardiothoracic surgery department had the highest rate (75.7%) of pRBC dispatch among patients who had ABO/Rh type and screening testing done, whereas the orthopedic surgery department had the lowest rate (15.1%). Furthermore, all surgical departments showed a crossmatch to transfusion ratio of less than 2.00.

Figure 1 shows the incidences of ABO/Rh type and screening at each year from 2006 to 2010. Compared to 2007, there was a significant reduction in the incidence of ABO/Rh type and screening and the incidence of pRBC requests among patients tested that persisted until 2010. The data from the ten most frequently performed surgeries overall are in Table 2. Table 3 shows the variability in ABO/Rh type and screening and transfusion practice within the orthopedic surgery department among different types of surgery.

## 4. Discussion

Our study shows the incidence of ABO/Rh testing and transfusion in a single pediatric hospital during the past 4 years. Our study is the first study to evaluate the utilization pattern of ABO/Rh testing and transfusion in pediatric patients, analyzed by surgical departments and surgical procedures.

Previous studies have tried to demonstrate the mechanisms for improving pRBC inventory management through detailed analysis of RBC preparation and transfusion practices in pediatric surgical procedure [5] and to establish patient-specific preoperative RBC preparation guidelines. Our study identified room for improvement in the utilization of ABO/Rh testing and blood bank resources [1,3,6,7]. In several studies, the maximum surgical blood order schedule (MSBOS) has been used to determine preoperative blood orders for specific surgical procedures at specific institutions [8]. On the other hand, in this study, we evaluated the utility of ABO testing using the ratio of actual transfusion to ABO/Rh type and screening in pediatric patients. Our result correlates with the result of the study from Johns Hopkins using MSBOS; it may be helpful to build tools for efficient use of blood bank resources based on the two results in further studies [8].

The frequency of ABO/Rh type and screening and transfusion varied among surgical departments and among types of surgery. The variation reflects the nature of the surgical procedure and the patient population of each department [9]. Although the ophthalmology department had the lowest rate of ABO/Rh type and screening, the pRBC dispatch rate among patients tested was 22.8%, which is higher than that of the orthopedic surgery department, urology department, and otorhinolaryngology department. In terms of intradepartmental variation, the orthopedic surgery department not only had the lowest pRBC dispatch rate among patients tested for ABO/Rh, but the variation among different types of surgery was also extensive. Only 1.5% of ABO/Rh tested patients undergoing hand or foot surgery had pRBC dispatched, whereas 30.5% of ABO/Rh tested patients undergoing hip/femur surgery had pRBC dispatched. This may imply that ABO/Rh type and screening is routinely performed in all patients scheduled for surgery, requiring patients who are unlikely to require transfusion to undergo the unnecessary test. Because blood draw can be more stressful in pediatric patients, as mentioned in the introduction, presurgical ABO/Rh type and screening should be performed only in pediatric patients who are scheduled to receive an operation with high possibility of transfusion, such as ventricular septal defect (VSD) closure at the cardiothoracic surgery department shown in Table 2.

Additionally, as shown in Figure 1, the frequency of ABO/Rh type and screening, pRBC requests, and RBC dispatches slowly decreased over the study period. This may be due to accumulation of surgical experience leading to greater consistency/confidence in the surgical procedure. The overall pRBC dispatch in patients for whom pRBC was requested was relatively constant.

In the operating room, the availability of blood products in a timely fashion can be critical. There is no standard requirement/guideline as to how quickly pRBCs must be ready for transfusion after the initial request [10,11,12]. In our hospital, which usually used an abbreviated crossmatching system, it takes less than 30 min from the time of request to pRBC dispatch without a previous ABO/Rh type and screening test, and less than 15 min when an ABO/Rh type and screening test has already been performed. Some patients may not tolerate waiting the additional 15 min for pRBC, but the possibility of such an urgent situation is very rare in elective surgery. Moreover, most institutions have O/Rh negative blood prepared to respond to this urgent but rare situation [13].

There are a few limitations to consider when interpreting our data to other situations. First, due to limitations of extracting data from the electronic medical records, our study used the incidence of the pRBC dispatch instead of actual transfusion. Considering that there are situations in which pRBC is dispatched but not transfused, actual transfusion rates may be lower than the pRBC dispatch rate. Second, our study data come from a single pediatric hospital. Application of our data to other institutions requires caution, as other hospitals may have different patient populations, policies, and protocols.

Routinely performed testing may not be necessary for all surgical departments nor every type of surgery [14]. A tailored approach depending on the planned surgical procedure may decrease the use of ABO/Rh type and screening and also the consequent discomfort and costs.

## Reference

## Figures and Tables

**Figure 1 jcm-08-01372-f001:**
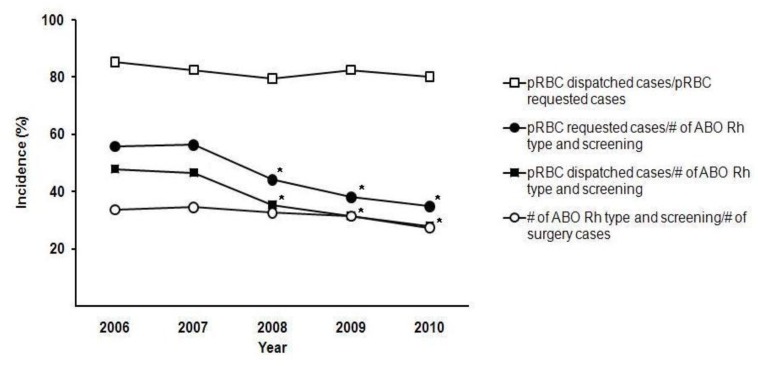
Temporal trend of ABO/Rh type and screening and transfusion. pRBC = packed red blood cell (* *p* < 0.05 compared to year 2007).

**Table 1 jcm-08-01372-t001:** Incidence of ABO/Rh type and screening and transfusion.

Surgical Department	No. of Surgery Case	No. of ABO/Rh Tests (%) ^*^		pRBC Requested Cases (%) ^§^		pRBC Dispatched Cases (%) ^†^		Crossmatch to Transfusion Ratio ^††^	pRBC Dispatched Cases among ABO/Rh Tested Cases (%)
Orthopedic surgery	6326	2886	(45.6)	689	(23.9)	435	(63.1)	1.58	15.1
Ophthalmology	5195	79	(1.5)	19	(24.1)	18	(94.7)	1.06	22.8
ENT	3718	168	(4.5)	33	(19.6)	26	(78.8)	1.27	15.5
Plastic surgery	3112	132	(4.2)	64	(48.5)	42	(65.6)	1.52	31.8
General surgery	3107	1077	(34.7)	548	(50.9)	386	(70.4)	1.42	35.8
Cardiothoracic surgery	1795	1757	(97.9)	1366	(77.7)	1331	(97.4)	1.03	75.7
Urology	1792	189	(10.5)	51	(27.0)	32	(62.7)	1.59	16.9
Neurosurgery	1332	1237	(92.9)	719	(58.1)	596	(82.9)	1.21	48.2
Total	23,631	7602	(32.2)	3511	(46.2)	2882	(82.1)	1.24	37.9

pRBC = packed Red blood cell, ABO/Rh test = ABO and Rh type and screening, ENT = ear, nose, and throat surgery, ^*^ ABO/Rh tested cases among all cases, ^§^ pRBC requested cases among ABO/Rh tested cases, ^†^ pRBC dispatched cases among pRBC requested cases, **^††^** pRBC requested cases divided pRBC dispatched cases.

**Table 2 jcm-08-01372-t002:** The incidence of ABO/Rh type and screening and transfusion in ten most frequently performed surgeries.

Type of Surgery	No. of Surgery Case	No. of ABO/Rh Tests (%) ^*^		pRBC Requested Cases (%) ^§^		pRBC Dispatched Cases (%) ^†^		Crossmatch to Transfusion Ratio ^††^	pRBC Dispatched Cases among ABO/Rh Tested Cases (%)
EOM recession	1314	6	(0.5)	1	(16.7)	1	(100)	1	16.7
Hernia repair	1267	65	(5.1)	21	(32.3)	16	(76.2)	1.31	24.6
T-op and/or A-op	1191	48	(4.0)	3	(6.3)	3	(100)	1	6.3
Palatoplasty, Alveoloplasty, cheiloplasty	931	13	(1.4)	3	(23.1)	2	(66.7)	1.5	15.4
Femur/hip op	969	807	(83.3)	404	(50.1)	246	(60.9)	1.64	30.5
VTI	838	6	(0.7)	0		0		-	0
FB removal	791	619	(78.3)	74	(12.0)	36	(48.6)	2.06	5.8
Epiblepharon repair	637	2	(0.3)	0		0		-	0
Osteotomy	626	574	(91.7)	243	(42.3)	193	(79.4)	1.26	33.6
IO myomectomy	541	3	(0.6)	0		0		-	0
VSD closure	538	538	(100)	385	(71.6)	381	(99.0)	1.01	70.8

Ten most frequently performed surgeries were analyzed. pRBC, packed red blood cell; ABO/Rh test, ABO and Rh type and screening; EOM, extra ocular muscle; T-op, tonsillectomy; A-op, adenoidectomy; VTI, ventilating tube insertion; FB, foreign body; IO, inferior oblique; VSD, ventricular septal defect; NA, not applicable, * ABO/Rh tested cases among all cases, ^§^ pRBC requested cases among ABO/Rh tested cases, ^†^ pRBC dispatched cases among pRBC requested cases, **^††^** pRBC requested cases divided pRBC dispatched cases.

**Table 3 jcm-08-01372-t003:** Frequency of ABO/Rh type and screening and transfusion in orthopedic surgery.

Type of Orthopedic Surgery	No. of Surgery Cases	No. of ABO/Rh Tests (%) ^*^		pRBC Requested Cases (%) ^§^		pRBC Dispatched Cases (%) ^†^		Crossmatch to Transfusion Ratio ^††^	pRBC Dispatched Cases among ABO/Rh Tested Cases (%)
femur/hip/pelvis	969	807	(83.3)	404	(50.1)	246	(60.9)	1.64	30.5
FBR	791	619	(78.3)	74	(12.0)	36	(48.6)	2.06	5.8
osteotomy	626	574	(91.7)	243	(42.3)	193	(79.4)	1.26	33.6
Hand and foot	519	453	(87.3)	59	(13.0)	7	(11.9)	8.42	1.5
lower leg and knee	503	423	(84.1)	73	(17.3)	40	(54.8)	1.83	9.5
spine deformity correction	71	71	(100.0)	59	(83.1)	52	(88.1)	1.13	73.2
Overall	6326	2886	(45.6)	689	(23.9)	435	(63.1)	4.58	15.1

The 6 most common orthopedic surgical procedures were analyzed. Packed red blood cell dispatched cases among ABO/Rh type and screening tested cases varied between 5.8% and 73.2%. pRBC, packed red blood cell; ABO/Rh test, ABO and Rh type and screening; FBR, foreign body removal, * ABO/Rh tested cases among all cases, ^§^ pRBC requested cases among ABO/Rh tested cases, ^†^ pRBC dispatched cases among pRBC requested cases, ^††^ pRBC requested cases divided pRBC dispatched cases.

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
