# Peer review of "The Utility of ABO Testing in Pediatric Patients Undergoing Elective Surgery"

_jcm, 2019, doi:10.3390/jcm8091372_

Round 1
Reviewer 1 Report
This is an interesting article from the perspective of pediatric surgery. The wide variation in testing vs transfusing will be of interest to pediatric surgeons and blood bankers.
The introduction and possibly also the discussion needs to give more emphasis to the point that pre-surgical type and screening is done to ensure blood is available for emergency transfusion. the more usual top-up transfusions do not pre-surgery testing because that could be left for the patients who actually need a non-urgent transfusion. It might also therefore be useful to look at the list of surgical procedures where the ratio of pRBC dispatched among ABO/Rh tested was very low, as this is the group where changes in policy would be able to take effect safely.
Related to the previous paragraph, it would also be helpful to know if the blood bank uses electronic crossmatching, abbreviated crossmatching ('immediate spin') or requires full serological crossmatching before issuing blood. The speed with which a blood bank can turn around an urgent request from a new sample is an important factor in deciding whether a pre-surgical sample is needed. These points are similar to the issues around MBOS (maximum blood ordering schedule) guidelines - balancing pre-surgical crossmatching vs urgent testing vs using O negative units.
This article could be even more useful if some guidance were given about when testing was/was not appropriate. This could take the form of a suggested cut-off (proportion of patients transfused), or a scoring system or other criteria based system, though this would require further data (e.g. patient demographics).
Lastly, the references are well presented but with a few minor errors (Rheineck-Leyssius' surame in reference 1 [correctly done in ref 6]; no doi for ref 9, incorrect presentation of group name in ref 12. References 8-13 do not appear in the text. I also wondered if there was any literature more recent than 2012 that might apply. doi:10.1111/pan.12250 is an example.
Author Response
Point 1: The introduction and possibly also the discussion needs to give more emphasis to the point that pre-surgical type and screening is done to ensure blood is available for emergency transfusion. the more usual top-up transfusions do not pre-surgery testing because that could be left for the patients who actually need a non-urgent transfusion. It might also therefore be useful to look at the list of surgical procedures where the ratio of pRBC dispatched among ABO/Rh tested was very low, as this is the group where changes in policy would be able to take effect safely.
Response 1: As you advised, it is necessary especially to perform pre-surgical type and screening before surgery of the high possibility of emergency transfusion for efficient use of limited blood. This is mentioned in the 3rd paragraph of the introduction of this study and in the 3rd paragraph of discussion.
Point 2: Related to the previous paragraph, it would also be helpful to know if the blood bank uses electronic crossmatching, abbreviated crossmatching ('immediate spin') or requires full serological crossmatching before issuing blood. The speed with which a blood bank can turn around an urgent request from a new sample is an important factor in deciding whether a pre-surgical sample is needed. These points are similar to the issues around MBOS (maximum blood ordering schedule) guidelines - balancing pre-surgical crossmatching vs urgent testing vs using O negative units.
Response 2: In our hospital, we do not use electronic crossmathing but we use abbreviated crosmatching. This will take 30 minutes from the time of request to pRBC dispatch without a previous ABO/Rh type and screening test and less than 15 minutes when ABO/Rh type and screening test has already been performed. This time, as you said, can be a critical factor in urgently needed blood transfusions in our hospital. Therefore, We have added the 5th paragraph of the discussion about the use of our abbreviated crossmatching system.
Point 3: This article could be even more useful if some guidance were given about when testing was/was not appropriate. This could take the form of a suggested cut-off (proportion of patients transfused), or a scoring system or other criteria based system, though this would require further data (e.g. patient demographics).
Response 3: Thank you for your good feedback. Based on our results, we were able to identify the trends in which departments would not be needed for ABO/Rh type and screening, and also in which of the top ten surgeries performed in our hospital. However, the possibility of transfusion may be required in different countries or hospitals, so multi-center studies are needed. Therefore, further studies are needed to develop a scoring system or criteria for ABO/Rh type and screening test.
Reviewer 2 Report
The manuscript by Kim et al. is well organized and clearly written. I think the manuscript would benefit from a discussion of Maximal Surgical Blood Order Schedules (MSBOS), which are designed to guide clinicians in their pre-surgical transfusion orders. Many articles have been written by Steve Frank from Johns Hopkins on this topic and should be included.
The second point is the crossmatch to transfusion ratio, which is a metric many blood banks use to measure their efficiency. Has this been calculated for the lab or for the data that has been pulled? I think it should be included in this work.
Also, does your hospital used electronic crossmatch to release samples to patient with a negative antibody screen? If so, this should be mentioned, as many transfusion services that use electronic XM no longer recommend type and screens for patients with historically negative screens and minimal risk of blood loss.
Author Response
Point 1: The manuscript by Kim et al. is well organized and clearly written. I think the manuscript would benefit from a discussion of Maximal Surgical Blood Order Schedules (MSBOS), which are designed to guide clinicians in their pre-surgical transfusion orders. Many articles have been written by Steve Frank from Johns Hopkins on this topic and should be included.
Response 1: Thank you for the good advice. In my opinion, the Maximal Surgical Blood Order Schedules (MSBOS) and our utility of ABO/Rh type and screening seem to be correlated, and we can develop a more objective tool for blood use furthermore. Also this may be particularly useful in paediatric patients who have limited preoperative testing. This is mentioned in the 2nd paragraph of discussion.
Point 2: The second point is the crossmatch to transfusion ratio, which is a metric many blood banks use to measure their efficiency. Has this been calculated for the lab or for the data that has been pulled? I think it should be included in this work.
Response 2: We've added crossmatch to transfusion ratio to the Result and Tables. In our hospital, we perform crossmatch in all cases when pRBC requested. So in the Tables of the study, when the requested pRBC is divided by the dispatched pRBC, the crossmatch to transfusion ratio can be known. In general, as I know, the appropriate Crossmatch to transfusion ratio is 2:1. In our cases, the ratios are less than or equal to 2:1 except for hand and foot orthopaedic surgery. In the case of hand and foot orthopaedic surgery, it may be beneficial to reduce the rate of crossmatch testing.
Point 3: Also, does your hospital used electronic crossmatch to release samples to patient with a negative antibody screen? If so, this should be mentioned, as many transfusion services that use electronic XM no longer recommend type and screens for patients with historically negative screens and minimal risk of blood loss..
Response 3: Thanks for mentioning the concept of electronic crossmatch. The electronic crossmatch you mentioned is very efficient for negative antibody patients, but it is not implemented in our hospital system. The system is also expected to be of great help in paediatric patients with blood collection difficulties.